# Strengthening health system's capacity for linkage to HIV care for adolescent girls and young women and adolescent boys and young men in South Africa (SheS'Cap-Linkage): Protocol for a mixed methods study in KwaZulu-Natal, South Africa

Edward Nicol [1,2]*, Wisdom Basera [1,3], Carl Lombard [4], Kim Jonas [5], Trisha Ramraj [6], Darshini Govindasamy [5], Mbuzeleni Hlongwa [1,7], Tracy McClinton-Appollis [5], Vuyelwa Mehlomakulu [1], Nuha Naqvi [8], Jason Bedford [8], Jennifer Drummond [8], Mireille Cheyip [8], Sibongile Dladla [8], Desiree Pass [1], Noluntu Funani [1], Cathy Mathews [5]

**1** Burden of Disease Research Unit, South African Medical Research Council, Cape Town, South Africa, **2** Division of Health Systems and Public Health, Stellenbosch University, Cape Town, South Africa, **3** School of Public Health and Family Medicine, University of Cape Town, Cape Town, South Africa, **4** Biostatistics Unit, South African Medical Research Council, Cape Town, South Africa, **5** Health System Research Unit, South African Medical Research Council, Cape Town, South Africa, **6** HIV Prevention Research Unit, South African Medical Research Council, Cape Town, South Africa, **7** School of Nursing and Public Health, University of KwaZulu-Natal, Durban, South Africa, **8** Division of Global HIV & TB, Centers for Disease Control and Prevention, Pretoria, South Africa

* Edward.Nicol@mrc.ac.za

## Abstract

### Introduction

Adolescent girls and young women (AGYW) aged 15–24 years and adolescent boys and young men (ABYM) aged 15–34 years represent one of the populations at highest risk for HIV-infection in South Africa. The National Department of Health adopted the universal test and treat (UTT) strategy in 2016, resulting in increases in same-day antiretroviral therapy initiations and linkage to care. Monitoring progress towards attainment of South Africa's 95-95-95 targets amongst AGYW and ABYM relies on high quality data to identify and address gaps in linkage to care. The aim of this study is to describe the current approaches for engaging AGYW and ABYM in the treatment continuum to generate knowledge that can guide efforts to improve linkage to, and retention in, HIV care among these populations in KwaZulu-Natal, South Africa.

### Methods and analysis

This is a mixed methods study, which will be conducted in uMgungundlovu district of Kwa-Zulu-Natal, over a 24-month period, in 22 purposively selected HIV testing and treatment service delivery points (SDPs). For the quantitative component, a sample of 1100 AGYW aged 15–24 years and ABYM aged 15–35 years old will be recruited into the study, in

**Data Availability Statement:** No datasets were generated or analysed during the current study. All relevant data from this study will be made available upon study completion.

**Funding:** This work will be funded by the South African Medical Research Council and the U.S. President's Emergency Plan for AIDS Relief (PEPFAR) through the Centers for Disease Control and Prevention, under the terms of Cooperative Agreement Number 1 NU2GGH002193-01-00, awarded to Edward Nicol and Cathy Mathews. https://www.cdc.gov/ The funders will not have a role in study design, data collection and analysis, decision to publish, or preparation of the manuscript.

**Competing interests:** The authors declare no competing interest.

addition to 231 healthcare providers (HCPs) involved in the implementation of the UTT program. The qualitative component will include 30 participating patients who were successfully linked to care, 30 who were not, and 30 who have never tested for HIV. Key informant interviews will also be conducted with 24 HCPs. Logistic regression will be used to model the primary outcomes on SDP types, while a time to event analysis will be conducted using a Cox regression model and adjusting the standard errors of the hazard ratio for the clustering of participants within SDPs. For qualitative data, a general inductive approach of analysis will be used.

## Dissemination

Findings from the study will be communicated to the study population and results will be presented to stakeholders and at appropriate local and international conferences. Outputs will also include a policy brief, peer reviewed journal articles and research capacity building through research degrees.

## Introduction

South Africa has the biggest HIV epidemic globally, with more than 7.8 million people living with HIV (PLHIV) in 2020 [1, 2]. Adolescent girls and young women (AGYW) aged 15–24 years represent one of the populations at highest risk for HIV-infection with an estimated HIV incidence of 1.5% [3]. Recent estimates show that 5.8% of adolescent girls aged 15–19 years were HIV-positive, compared with 4.7% of adolescent boys in that age group in 2017. In the 20–24 year age group, 10.9% of young women were HIV-positive compared with 4.8% of young men, while in the 25–29 and 30–34 year age groups, 27.5% and 34.7% of women were HIV-positive compared to 12.4% and 18.4% of men, respectively [3]. These figures highlight the specific vulnerability of AGYW, but they also show that adolescent boys and young men (ABYM) have a substantial risk of HIV.

Despite South Africa's remarkable progress with antiretroviral therapy (ART) scale-up, ART coverage remains sub-optimal, particularly among HIV-positive AGYW and ABYM aged 15–24 years of whom only 55% of ABYM and 52% of AGYW were receiving ART in 2022 [4, 5]. Among people living with HIV aged 25–49 years, an estimated 63.1% were receiving ART, showing that there is a gap in linkage to, and retention in, HIV treatment services. Less than 70% of HIV-positive adolescents and young people aged 15–24 years were virally suppressed in 2022 [5]. These data highlight the challenges in linking and retaining adolescents and young people in HIV care, which is a major public health challenge for the South African National Department of Health.

Several factors account for the increased vulnerability of AGYW to HIV infection including intergenerational sexual relationships, gender power imbalances, gender-based violence, poverty and the low status of women, exclusion from economic opportunities and limited access to secondary schooling [6, 7]. Relationships with older men lead to power imbalances increasing the likelihood of intimate partner violence and the non-use of condoms during sex [7]. Socially excluded, marginalised adolescents and young people are particularly vulnerable to HIV infection and have poor access to HIV treatment and care, including people with disabilities, people who use drugs (PWUD), lesbian, gay, bisexual, transgender, and intersex (LGBTI) people, sex workers and undocumented migrants.

While AGYW are disproportionately affected by HIV, heterosexual men remain a critical population in HIV prevention. Men are less likely than women to test for HIV, engage in care in a timely way and remain in care [8–10]. In South Africa in 2018, 93% of women living with HIV were aware of their status compared to 88% of HIV-positive men [11]. Programs for the prevention of mother to child transmission of HIV enable women to access HIV testing services during routine antenatal appointments [10–12], which partly accounts for more women testing for HIV compared to men. Men report worrying that queuing outside a testing facility will be taken as evidence that they are living with HIV and avoid testing because they are terrified of a positive result [13]. While women have the benefit of a woman-centric health system, hegemonic masculinities largely exclude men from engaging with the health care systems [10]. HIV stigma, homophobia and gender norms also threaten men's engagement in HIV prevention and treatment services [10].

As per the Joint United Nations Program on HIV and AIDS (UNAIDS) and the World Health Organisation (WHO) HIV strategic recommendations for countries, South Africa embarked on rolling out Universal Test and Treat (UTT) in September 2016. UTT is a WHO-supported initiative believed to be key for attaining the UNAIDS fast-track targets, aimed at reducing both HIV mortality and new infections to below 500 000 by 2020 [14]. The 2030 targets included the 95-95-95 targets, that state that 95% of PLHIV should know their status, 95% of those who know their status should be on treatment and 95% of those on treatment should be virally supressed [15]. UTT advocates that all individuals testing for HIV be initiated on treatment regardless of their clinical staging or CD4 count [16]. Through UTT, HIV transmission can be reduced by early initiation to ART coupled with adherence to treatment. This suppresses the viral load in infected persons, thus reducing the risk of transmission [17, 18]. To attain the potential benefits of UTT strategies, those who test HIV-positive must be linked to care for treatment initiation immediately after testing, regardless of where they tested [19]. Further, in order to achieve viral load suppression, all those on ART must be retained in care and adhere to lifelong ART [20].

Monitoring progress towards attainment of South Africa's 95-95-95 strategic HIV treatment targets relies on high quality data to identify and address gaps in ART coverage, quality of care, prescription adherence and resource allocation. This study, which will monitor linkage to, and retention in HIV care among AGYW and ABYM in the uMgungundlovu district, KwaZulu-Natal, seeks to answer three questions. Firstly, what are the available service delivery models that are seeing higher linkage to care, and continuity of care and treatment among AGYW and ABYM? Secondly, what is the rate of acceptability and uptake of linkage to care by service delivery models–facility-only, community-only, and community-facility hybrid models? Thirdly, what factors influence acceptability and decision-making around the use of different service delivery models? The study aims to describe the current approaches used to engage AGYW and ABYM in the treatment continuum to generate knowledge which can be used to provide recommendations on the best practices for linkage to and retention in HIV care services for AGYW age 15–24 years and ABYM aged 15–35 years.

## Materials and methods

### The objectives of the study are

1. To describe the socio-demographic characteristics of AGYW (15–24 years) and ABYM (15–35 years) newly diagnosed with HIV who access HIV testing services in the uMgungundlovu district of KwaZulu-Natal, and to compare service delivery models.

2. To describe the HIV testing experiences of AGYW and ABYM newly diagnosed with HIV in the uMgungundlovu district, comparing experiences across service delivery models.

3. Among HIV-positive AGYW and ABYM, to quantify linkage to and retention in HIV care across different service delivery models, and to describe the factors associated with linkage to and retention in HIV care and health impacts thereof, in the uMgungundlovu district.

4. To identify the mechanisms (reasoning and decision-making) and context (individual, structural social and health systems factors) that influence AGYW's and ABYM's uptake of HIV treatment and care services to describe the relative success of different HIV treatment service delivery models.

## Setting

The setting for this study, which will be conducted in the uMgungundlovu district in Kwa-Zulu-Natal, have been described extensively in a complimentary study [21]. This predominantly rural district, with high HIV prevalence of 24% and 37% for people aged between 15–49 years among males and females, respectively [22], an 89% medically uninsured population, offers an excellent setting for monitoring UTT alongside interventions seeking to improve linkage to HIV care in the public sector. uMgungundlovu district is comprised of seven municipalities, all of which have rural areas, namely uMshwathi, Umgeni, Mpofana, Impendle, Msunduzi, uMkhambathini and Richmond.

## Study design

The SHeS'Cap-Linkage study, which compliments the SHeS'Cap-PrEP study [21], is a sequential, explanatory, mixed methods study that will be conducted at 22 purposively selected HIV testing and treatment service delivery points (SDPs). Unlike the SHeS'Cap-PrEP study [21] that focused on AGYW and ABYM who tested HIV negative at the selected pre-exposure Prophylaxis (PrEP) SDPs (health clinics, school-based and community-based centers), this study will focus on AGYW and ABYM with a positive HIV test outcome. These will include quantitative analysis of routine program monitoring data; monitoring of linkage to and retention in HIV care among AGYW and ABYM who test HIV-positive in selected SDPs, and in-depth interviews (IDIs) with AGYW and ABYM to describe the barriers to, and enablers of linkage to, and retention in HIV care. We will also conduct key informant interviews (KIIs) with healthcare providers involved in the implementation of the UTT program in the district.

The primary outcomes for this study are linkage to care within 14 days and retention in care at six months after HIV diagnosis. Linkage to care is defined as "the proportion of AGYW (15–24 years) and ABYM (15–35 years) per SDP per month who have been initiated onto ART as evidence by a record in their TIER.Net or for whom baseline CD4 results have been captured into their TIER.Net record within 0–14 days (within 2 weeks) of their HIV-positive test at enrolment." Retention in care is defined as "the proportion of AGYW (15–24 years) and ABYM (15–35 years) per SDP per month for whom an entry has been captured into the National Health Laboratory Service (NHLS) for confirmed linkage at 6 months after their positive HIV test at enrolment."

## Study population and recruitment

The study population will consist of two groups: a.) healthcare providers, comprising of district and health facility managers, nurses, health information officers, staff involved in data collection at facility and district levels, program managers, monitoring and evaluation officers responsible for each of the seven sub-districts (strata) in the uMgungundlovu district, and b.) AGYW aged 15–24 years and ABYM aged 15–35 years old who test HIV-positive during the

**Table 1. Eligibility criteria for the SHeS'Cap-Linkage study in uMgungundlovu district, KwaZulu-Natal, South Africa, 2021–23.**

| Inclusion criteria | Exclusion criteria |
|---|---|
| • Have accepted an HIV test in one of the participating service delivery points from July 2021 to June 2022 | • Test HIV-negative on the day of recruitment. |
| | • No access to a cellphone |
| • Have access to a cell phone and willingness to provide contact details (required for follow up). | • Unwilling to provide contact details |
| | • Adolescents <15 years of age |
| • Be 15–24 years old for AGYW, or 15–35 years old for ABYM. | • Young women over 24 years of age, and men over 35 years of age |
| • Be seropositive based on HIV rapid test results on the day of recruitment. | • SDPs that don't provide HIV services |
| • Be able and willing to provide written informed consent. | • Healthcare providers not involved in implementing the Universal Test and Treat program |
| • SDPs providing HIV services | |
| • Healthcare providers involved in implementing the Universal Test and Treat program in selected SDPs | |

routine facility-based, school-based, and community-based HIV testing services. This cohort will be recruited and followed up at one month using questionnaires and at 6 months through routine data. Eligible participants will be considered based on the inclusion and exclusion criteria listed in Table 1.

Participants will be enrolled in the study as they await HIV testing in participating primary health care health facilities, schools and community-level HIV testing and treatment points (described collectively as SDPs). Upon enrolment, they will be grouped into two distinct cohorts for analysis:

- Newly enrolled AGYW and ABYM who test HIV-positive in a participating SDP on the day of enrolment during the 12 months data collection period (current protocol).

- Newly enrolled AGYW and ABYM who test HIV negative on the day of enrolment (SHeS'-Cap-PrEP cohort) [21].

Participants will be followed up at 1 and 6 months after enrolment using quantitative and qualitative data collection methods. Data on the HIV testing experience of AGYW and ABYM will be collected cross-sectionally, whereas the linkage and retention data will be monitored prospectively over six months using routine data. In addition, health service and system contextual factors that may influence linkage to and retention in HIV care in participating SDPs will be investigated. Data collection will be over a 24-month period (August 2021 to July 2023). AGYW and ABYM will be recruited at selected SDPs and followed up for 7 months using quantitative and qualitative data collection methods.

## Sampling

**Service delivery points (SDPs) and sample size for individuals to be enrolled.** The criteria for selecting the 22 SDPs included in this study have been reported elsewhere [21], and the HIV testing number in these SDPs are >4000 clients per month. Published studies and previous surveillance data from KwaZulu-Natal province in South Africa demonstrate an average linkage to care rate of 62% post HIV testing. However, recent findings from a district neighboring uMgungundlovu, reported a linkage to care rate of 83% [23]. We assume the linkage to care rates in uMgungundlovu district municipality to be 10% higher than 62% reported from the province based on the possible impact of UTT on HIV uptake. Also, we intend to compare the three service delivery models (facility-based, school-based, and community-based) across

**Table 2. Power and sample size for the comparison of service delivery model for the SHeS'Cap-Linkage study in uMgungundlovu district, KwaZulu-Natal, South Africa, 2021–2023.**

| alpha | Beta | $k_1$ | $k_2$ | $m_1$ | $m_2$ | delta | $p_1$ | $p_2$ | rho |
|-------|------|-------|-------|-------|-------|-------|-------|-------|-----|
| 0.05 | 0.89 | 11 | 11 | 50 | 50 | 0.1 | 0.62 | 0.72 | 0.01 |

Where alpha—type 1 error; beta—power; m–average cluster size; k—number of clusters; delta–difference between proportions; p–linkage to care estimate (proportion); rho–inter-cluster correlation.

the seven sub-districts in uMgungundlovu. Since the study is non-randomised (unmatched study) with purposive sampling, it will be exploratory in nature and will inform the rates of linkage to care based on the different SDPs to facilitate better planning for intervention studies.

Sample size calculations (Table 2) are illustrative of the power of the study design to detect difference in coverage by service delivery models and are based on the comparison of proportions between clusters of fixed sizes in each service delivery group at a specific time point (i.e., 6 months follow-up). The sample size was done for the comparison of mainly community versus facility service delivery models for all the participants with the assumption that the participants attending schools i.e., TVET colleges and High schools will be included in the facility and community-based arms. The number of clusters (SDPs) needed to detect a coverage difference between the community- and facility-based service delivery models with >80% power is 22 in total, translating to 11 in each model with 50 participants per cluster assuming an intraclass correlation coefficient (ICC) of 0.01 and significance level of 0.05 [24]. The sample and power calculation were done using Stata v14.2 [25]. The study will retain at least 80% power for a 20% dropout at 6 months of follow-up (m = 40 per cluster) and the design effect of 1.5 were deemed realistic for the study populations.

The quantitative component of the study will target to enrol from the available 22 service delivery points a sample size of 1100 (50 participants x 22 SDPs) across all three primary service delivery models during the 24 months study period. Since there is a small number of clusters the comparison of the service delivery models will be done at the cluster level and between the SDPs since an analysis at the participant level is not advised. For the qualitative component, 90 purposively selected participants, aggregated by age and sex, who had enrolled at one of the SDPs and a fraction of those who have never had an HIV test will be interviewed. Based on the 1-month follow-up outcome data, 30 participants who were successfully linked to care, 30 who did not and 30 who have never tested for HIV will be included.

Furthermore, to investigate the health service and system contextual factors that may influence linkage to and retention in HIV care in participating SDPs, a purposive sample of 231 healthcare providers will be selected based on their experience implementing the UTT program (Table 3). These will consist of at least one from each available cadres per participating SDP and staff from each of the seven local municipalities offices, including the district office.

## Data collection

Baseline quantitative data collection will commence simultaneously in all 22 participating SDPs using self-administered electronic questionnaires built into REDCap [26]. The enrolment and data collection processes are summarized in Fig 1.

Secondary data sources, including routine data from facility-based health information systems, routine data from district-level TIER.Net, data from the National Health Laboratory Service (NHLS) and data from the South African Medical Research Council (SAMRC)'s Rapid Mortality Surveillance (RMS) containing monthly information about deaths registered by the

**Table 3. Sampling strategy for selecting healthcare providers across the SDPs for the SHeS'Cap-Linkage study in uMgungundlovu district, KwaZulu-Natal, South Africa, 2021–2023.**

| Function | SDP Level | Sub-district Level | District Level | Total staff to be interviewed |
|---|---|---|---|---|
| CEO/ Facility managers* | 12 | | | 12 |
| Linkage officer | 44 | | | 24 |
| HIV/lay counsellor | 44 | | | 24 |
| Professional Nurse* | 12 | | | 12 |
| Enrolled Nurse* | 12 | | | 12 |
| Community health worker | 24 | | | 24 |
| Facility information officer* | 12 | | | 12 |
| Data capturer* | 24 | | 4 | 28 |
| HAST Coordinator | | 7 | | 7 |
| PHC Manager | | 7 | | 7 |
| Sub-district manager | | 7 | | 7 |
| District information officer | | | 1 | 1 |
| District manager | | | 1 | 1 |
| **Total** | **204** | **21** | **6** | **231** |

*Primary health care (**PHC**) facilities linked to all 22 SDPs; **CEO**: Chief executive officer; **HAST**: HIV and AIDS / STI / TB.

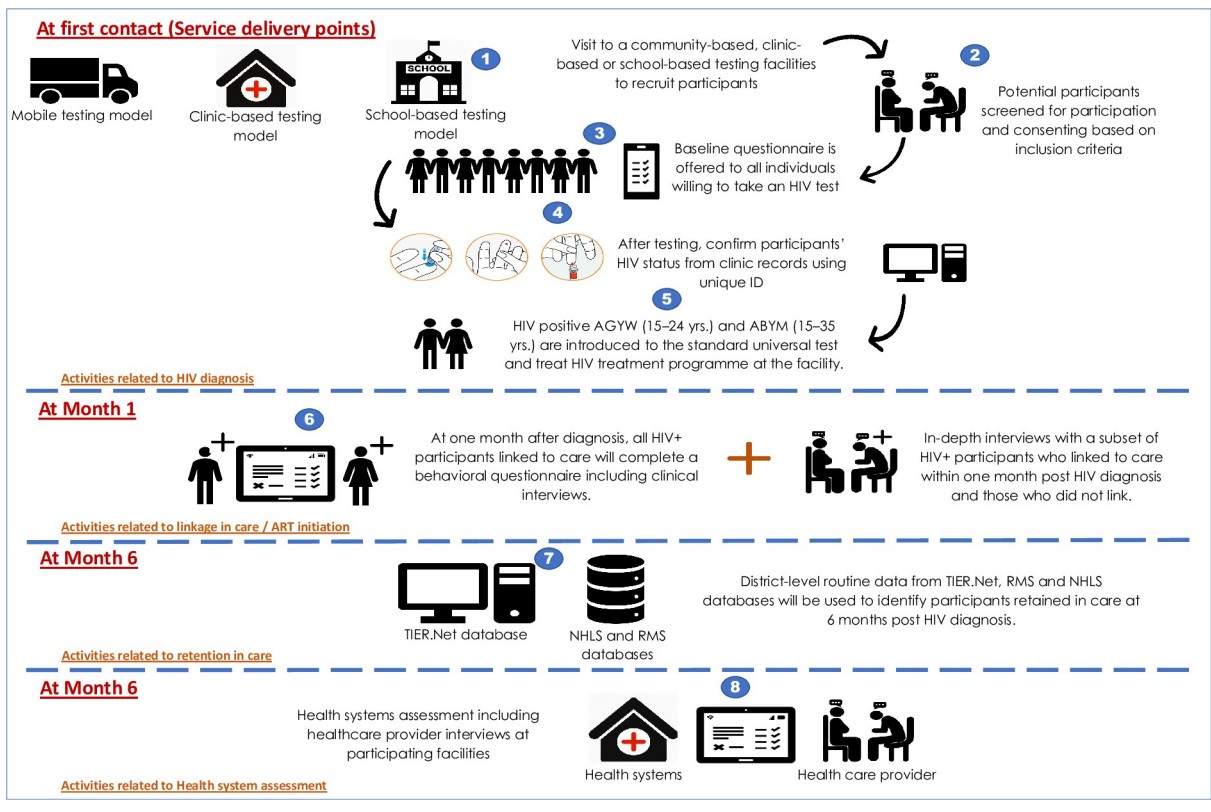

**Fig 1. An illustration of the study activities for the SHeS'Cap-Linkage study in uMgungundlovu district, KwaZulu-Natal, 2022.**

**Table 4. Roles of outcome variables in the SHeS'Cap-Linkage study in uMgungundlovu district, KwaZulu-Natal, 2021–2023.**

| Outcome variables | Examples of variables | Data sources and tools |
|---|---|---|
| • Proportion of AGYW and ABYM who have initiated ART as evidenced by a record in their TIER.Net within 14 days or for whom baseline CD4 results have been entered into their TIER.Net record within 1 month of their HIV-positive test result at enrollment. (Primary outcome) | • Patient receipt of HIV test result. <br> • Patient attendance at first clinic appointment after receipt of HIV result. <br> • Enrolled patients taking up same-day initiation onto ART at diagnosis <br> • Proportion for whom ART initiation is within 2 weeks of a CD4 test <br> • ART initiation date recorded in clinical stationery and/or TIER.Net | • Routine data (Clinical charts, clinic HIV testing and tracking registers, patient folders and TIER.Net*) and questionnaire data as necessary. |
| • Proportion of AGYW and ABYM for whom an entry has been captured into their TIER.Net record at 6 months after their positive HIV test at enrollment. (Primary outcome) | • Patient attendance at clinic as per contemporary NDoH guidelines at 6 months after HIV diagnosis | • Routine data (Clinical charts, clinic HIV testing and tracking registers, patient folders and TIER.Net*) |
| | • NHLS records of samples received at NHLS laboratory, as per contemporary NDoH guidelines (e.g., repeat CD4 or Viral Load results every 6 months) | • NHLS records <br> • RMS <br> • Baseline and follow-up questionnaires |
| | • TIER.Net records at facility of NHLS results recorded within 1 month of the date each sample was sent <br> • Viral load suppression at 6 months | |
| • Proportion of enrolled participants testing HIV-positive and not yet initiating ART who are not retained in HIV care at 6 months after HIV diagnosis at initiation | | • Routine data (Clinical charts, clinic HIV testing and tracking registers, patient folders and TIER.Net*) |
| • Further description of participants' health seeking behaviour | • Participant experiences of seeking HIV testing and care prior to date of HIV diagnosis at enrolment in this cohort. <br> • Proportion of patients who are not retained in care after initiating ART | • In-depth interviews (IDIs) |
| • Clinic processes and data quality to support linkage to and retention in care. | • Completeness and location of post-test initiation (clinic/community/schools) | • Routine data (Clinical charts, clinic HIV testing and tracking registers, patient folders and TIER.Net*) |
| • Association between SDPs processes and linkage to and retention in care and psychosocial factors including social support and age-appropriate disclosure, health literacy and beliefs, HIV stigma and expectations of health services | • Proportion initiating ART immediately after a positive HIV test at SDPs | • Baseline and follow-up questionnaires |

*TIER.Net is an electronic patient management system used to capture patient-level data on HIV management in South Africa. **NHLS:** National health laboratory system; **RMS:** Rapid Mortality Surveillance; **AGYW:** Adolescent girls and young women; **ABYM:** Adolescent boys and young men; **ART:** Antiretroviral therapy; **SDPs:** Service delivery points.

South African Department of Home Affairs will be used to obtain information on linkage to and retention in care in the SDPs where this project will be implemented. The RMS, therefore, will be used to confirm that the individual considered as not linked to care or not retained in care was not deceased. Table 4 gives an overview of the outcome variables, and data sources and tools.

A health system/facility assessment survey tool [27, 28] will be used to collect both quantitative and qualitative data. The qualitative data will focus specifically on healthcare providers' perceptions on the implementation of UTT. Healthcare providers at each SDP will be approached by interviewers and depending on availability, an appointment will be secured. This is necessary not to disrupt the daily functioning of the SDPs. In addition, HAST and HIV service managers in the district health team will also be invited to contribute to the semi-

structured interviews. Given this study's emphasis on strengthening the quality and use of routine data, we will use the Performance of Routine Information Management (PRISM) framework [29] to structure our health system assessments. An example of how the PRISM framework and tools [30] have been customised in a particular setting by the lead author is described elsewhere [31]. The health system assessment tool has been field tested, refined and used in another study [23].

Health system readiness data relevant to the following domains will be collected. Domains include readiness to implement new policies and systems, leadership and governance, clinical processes (e.g., package of care delivered, integration with TB and other services, monitoring and evaluation/ information management, clinical oversight), drug supply and support services management (e.g., laboratory services), human resource management, and financial management.

We will also collect qualitative data using IDIs and KIIs. IDIs will be undertaken with a subset of 90 enrolled male and female participants, to contribute to a nuanced understanding of the drivers and dynamics of linkage to and retention in HIV care, including the mechanisms (reasoning and decision-making) and context (individual, structural social and health systems factors) that influence AGYW's and ABYM's uptake of HIV treatment and care services to describe the relative success of different HIV treatment service delivery models. In order to obtain wide-ranging responses, a maximum variation purposive sampling technique will be applied to include a varied representation of participants in terms of age, gender, marital status, employment status, geographic area and location of SDPs utilized. These will include:

- At least one enrolled participant from each participating SDP

- Participants who tested HIV-positive at participating SDPs who did and did not link to care within 14 days, and who did and did not remain in care at 6 months after their diagnosis, if non-linked participants can be reached

- Participants who enrolled in the study and responded to the baseline interview but did not test for HIV.

Furthermore, KIIs will be conducted with 22 purposively selected stakeholders including reproductive and healthcare providers at different sites and primary health care centers. The KIIs will focus on the barriers to, and facilitators of linkage to care, and perspectives of ways to improve the uptake of HIV treatment and care services to describe the relative success of different HIV treatment service delivery models. During the KIIs, process mapping will also be conducted to understand the processes and steps involved in the delivery of the services in each of the models. In this case, emphasis will be placed on the inputs, outputs, and outcomes. The duration of the IDIs, and KIIs will range from 60–120 minutes depending on how the discursive and interactive process unfolds.

## Data management

Fieldwork supervisors will do daily quality checks on completeness of REDCap records on the tablets, before uploading data over 3G or Wi-Fi to the REDCap folder stored securely on the SAMRC's server. A detailed description of the data management plan has been described elsewhere [21].

## Data analysis

The number of enrolled participants per SDPs over time and the number for whom 1- and 6-month outcome data are available will be presented. Throughout the analysis, outcomes will

be disaggregated by HIV status and gender, which will be a covariate in analysis where sample size permits.

Analysis of 1-month and 6-month data (from questionnaires and routine data sources) will guide stratification and reporting by additional outcomes, such as proportion who took up the offer of immediate ART initiation at HIV diagnosis. Baseline characteristics of enrolled participants will be summarized by pre-/post-exposure status, to allow consideration of selection biases and lack of balance. The primary outcome will be modelled on SPD type using a logistic regression model and the standard errors of the estimated odds ratios will be adjusted for the clustering of participants within SDPs. Given the small number of clusters the degrees of freedom will be adjusted using the Between-Within method recommended by Li & Redden [32]. A time to event analysis will also be conducted for the time to ART initiation from the date of testing HIV positive, using a Cox regression model and adjusting the standard errors of the hazard ratio for the clustering of participants within SDPs. Demographic variables such as age and gender as well as barriers and facilitators of linkage to care will also be considered in the modelling process.

The process evaluation will be analysed through mixed methods. Linkage to care and retention in care will be expressed as proportions of the HIV-positive cohort. Characteristics of barriers and enablers to linkage to or retention in care will also be expressed as proportions and stratified by age and gender. A p-value of $< 0.05$ will be considered statistically significant. A table shell is presented in S1 and S2 Tables.

Univariate analyses will also be used to describe the background characteristics of healthcare providers and health service readiness. Bivariate analysis will be done by cross-tabulating background characteristics of healthcare providers (age, education, job category, place of work, gender, and years employed) with the behavioural and organizational factors. Bar charts will be used to display bivariate analyses of key behavioural factors and organizational factors. A comparison between the different SDPs will be undertaken as well as an analysis to assess the effect of management support on the behavioural factors around health information. In addition to the above analyses, information from key informants on organisational determinants affecting the District Health Management Information Systems' performance will be collated.

A general inductive approach, driven by the main questions of interest (e.g., barriers and facilitators of linkage to care), will be used to analyze the qualitative data. Two co-investigators (including the lead author) will read each translated transcript independently to gain a general understanding of its content and scope, and thereafter will develop a structured coding framework guided by the main questions of interest, as well as allowing for new concepts to emerge through open coding. The independent analyses will then be compared for consistency; areas of discrepancies will be identified through critical evaluation of the sets of themes. The source quotes will be reviewed and agreed upon, and a final thematic report will then be generated from the combined analyses. Through comparative analysis within and between interviews, we will develop categories of coded information which will then be linked to broader themes emerging across interviews. The same codes will be used to analyse the data within ATLAS.ti software and this will be used to compare and complement the manual analysis. Data from the qualitative interviews will be triangulated, where appropriate, with quantitative data emerging from the questionnaires.

## Ethics and dissemination

Ethical approval for this study was obtained from the South African Medical Research Council Health Research Ethics Committee (Ref #: EC052-11/2020) on 19 January 2021. Gatekeeper

permissions were also obtained from the KwaZulu Natal Provincial Departments of Health (Ref #: KZ_202010_032), the uMgungundlovu health districts, and facilities. This project was also reviewed in accordance with CDC human research protection procedures and was determined to be research, but CDC investigators did not interact with human subjects or have access to identifiable data or specimens for research purposes. A waiver of parental consent has also been granted for participants aged 15–17 years, and documentation and informed consent process will be followed as for participants 18 years and older.

We will obtain oral and written informed consent from all potential participants in the study prior to their participation. Participation will be voluntary, and participants will be informed during recruitment that they could withdraw at any stage without consequences from the study team or the facilities which they attend. Any event deemed to be a serious adverse event (e.g., breach of participant confidentiality) will be systematically recorded and will follow the Standard Operating Procedures of the SAMRC Ethics Committee, which includes reporting adverse or unexpected events within 48 hours to the SAMRC Ethics Committee. Participation in the study may include disclosure of emotional and personal issues such as violence and stigma. Field staff will be trained on first-line response to disclosure of violence (LIVES) and will be capacitated to provide supportive referrals through the provision of a referral directory, should this be needed.

Dissemination of results will be extensive, both through scientific (publications and conference presentations) and through feedback and materials for the department of health, at facility, district, and national levels. The findings of the study will be presented to facility-based, district, provincial and national stakeholders, implementing partners, researchers in the scientific community and members of the public of South Africa, including participants attending primary care facilities in uMgungundlovu district. The anonymity and confidentiality of the participants will be preserved by not revealing any identifying information during the dissemination of study findings. Outputs will include a brief policy-relevant summary of findings, peer-reviewed journal articles, research capacity building through one or more research degrees, and materials to support enhanced capacity for scale-up of the interventions.

## Discussion

The aim of this study is to provide evidence to guide efforts to improve linkage to, and retention in HIV care among adolescent girls and boys, and young women and men in KwaZulu-Natal, in the context of the UTT strategy. Through UTT, HIV transmission can be reduced by early initiation to ART coupled with adherence to treatment. This suppresses serum concentration of the HIV virus (viral load count) in infected persons, reducing the risk of transmitting the virus to a negative person [17, 18, 33]. To attain the potential benefits of UTT strategies, those who test HIV-positive must be linked to care for treatment initiation immediately after testing, regardless of where they were tested [19]. Further, in order to achieve viral load suppression, all those on ART must be retained in care and adhere to lifelong ART [18]. This study has the potential to provide epidemiological data to further strengthen the HIV program in KwaZulu-Natal, and guide initiatives aimed at reaching the UNAIDS 95-95-95 targets.

The strengths of this study include the combination of research methods to answer the research questions. The study will combine data collected from people who take up HIV testing services and test HIV-positive with program monitoring through routine data to generate evidence to guide improvements in district and facility HIV services and ultimately to assist South Africa in reaching the UNAIDS 95-95-95 targets. The possible limitations of this study include the fact that we will be sampling adolescents which require the study team to be more sensitive and aware of their well-being during data collection as they might not be able to

adequately express their health care service experiences. However, field staff will be trained on sensitive interview techniques and the provision of LIVES, to ensure adolescents feel safe and are able to disclose personal information. Other limitations have been described extensively in a complimentary study [21].

## Supporting information

**S1 Table. Socio-Demographic characteristics of the HIV-positive participants at baseline and linkage in care in the first month of follow-up.**
(DOCX)

**S2 Table. Factors influencing linkage to care after the first month of HIV-positive diagnosis.**
(DOCX)

## Acknowledgments

**Disclaimer:** The findings and conclusions in this manuscript are those of the authors and do not necessarily represent the official position of the funding agencies.

## Author Contributions

**Conceptualization:** Edward Nicol, Darshini Govindasamy, Jennifer Drummond, Cathy Mathews.

**Data curation:** Edward Nicol, Wisdom Basera, Carl Lombard, Cathy Mathews.

**Formal analysis:** Edward Nicol, Wisdom Basera, Cathy Mathews.

**Funding acquisition:** Edward Nicol, Cathy Mathews.

**Investigation:** Edward Nicol, Wisdom Basera, Kim Jonas, Trisha Ramraj, Darshini Govindasamy, Mbuzeleni Hlongwa, Nuha Naqvi, Jason Bedford, Jennifer Drummond, Mireille Cheyip, Sibongile Dladla, Cathy Mathews.

**Methodology:** Edward Nicol, Wisdom Basera, Carl Lombard, Kim Jonas, Trisha Ramraj, Darshini Govindasamy, Nuha Naqvi, Jason Bedford, Jennifer Drummond, Mireille Cheyip, Sibongile Dladla, Cathy Mathews.

**Project administration:** Edward Nicol, Tracy McClinton-Appollis, Desiree Pass, Noluntu Funani, Cathy Mathews.

**Resources:** Edward Nicol, Cathy Mathews.

**Supervision:** Tracy McClinton-Appollis, Desiree Pass, Noluntu Funani, Cathy Mathews.

**Validation:** Edward Nicol, Carl Lombard, Kim Jonas, Nuha Naqvi, Jason Bedford, Jennifer Drummond, Mireille Cheyip, Sibongile Dladla, Cathy Mathews.

**Visualization:** Edward Nicol.

**Writing – original draft:** Edward Nicol.

**Writing – review & editing:** Edward Nicol, Wisdom Basera, Carl Lombard, Kim Jonas, Trisha Ramraj, Darshini Govindasamy, Mbuzeleni Hlongwa, Tracy McClinton-Appollis, Vuyelwa Mehlomakulu, Nuha Naqvi, Jason Bedford, Jennifer Drummond, Mireille Cheyip, Sibongile Dladla, Desiree Pass, Noluntu Funani, Cathy Mathews.

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
