## [Decision Letter · Decision Letter 0]

26 Sep 2022

PONE-D-22-19444Strengthening Health System’s Capacity for Linkage to HIV Care for adolescent girls and young women and adolescent boys and young men in South Africa (SheS’Cap-Linkage): Protocol for a mixed methods study in KwaZulu-Natal, South Africa.PLOS ONE

Dear Dr. Nicol,

Thank you for submitting your manuscript to PLOS ONE. After careful consideration, we feel that it has merit but does not fully meet PLOS ONE’s publication criteria as it currently stands. Therefore, we invite you to submit a revised version of the manuscript that addresses the points raised during the review process.

We look forward to receiving your revised manuscript.

Kind regards,

Sandra Boatemaa Kushitor, Ph.D.

Academic Editor

PLOS ONE

Journal Requirements:

" The funders had and will not have a role in study design, data collection and analysis, decision to publish, or preparation of the manuscript."

5. We note that Figure 1 in your submission contain map images which may be copyrighted. All PLOS content is published under the Creative Commons Attribution License (CC BY 4.0), which means that the manuscript, images, and Supporting Information files will be freely available online, and any third party is permitted to access, download, copy, distribute, and use these materials in any way, even commercially, with proper attribution. For these reasons, we cannot publish previously copyrighted maps or satellite images created using proprietary data, such as Google software (Google Maps, Street View, and Earth). For more information, see our copyright guidelines: http://journals.plos.org/plosone/s/licenses-and-copyright.

Additional Editor Comments:

Just as observed by the reviewers, the methods is confusing. The authors have to clarify exactly what they will be doing. I suggest the following

Abstract

- present all methods related to the quantitative study first. Present the qualitative methodology afterwards. Please dont mix up the two

- 'The questionnaire study population comprises of 231 healthcare providers and AGYW aged 15-24 years and ABYM aged

15-35 years old.' This is quite confusing because the authors mentioned that they will be interviewing 1100 participants.

Introduction

- the authors should include additional references. Ref 3 has been cited too often.

Methods

- About how many clients test for HIV at the clinics. The authors should specify, since this will influence the number the participants they will be able to recruit

Reviewers' comments:

Reviewer's Responses to Questions

**Comments to the Author**

1. Does the manuscript provide a valid rationale for the proposed study, with clearly identified and justified research questions?

Reviewer #1: Partly

Reviewer #2: Yes

2. Is the protocol technically sound and planned in a manner that will lead to a meaningful outcome and allow testing the stated hypotheses?

Reviewer #1: Yes

Reviewer #2: Partly

3. Is the methodology feasible and described in sufficient detail to allow the work to be replicable?

Reviewer #1: Yes

Reviewer #2: No

4. Have the authors described where all data underlying the findings will be made available when the study is complete?

Reviewer #1: No

Reviewer #2: Yes

5. Is the manuscript presented in an intelligible fashion and written in standard English?

Reviewer #1: Yes

Reviewer #2: Yes

6. Review Comments to the Author

You may also provide optional suggestions and comments to authors that they might find helpful in planning their study.

Reviewer #1: The authors have set out to outline the study protocol entitled Strengthening Health System’s Capacity for Linkage to HIV Care for adolescent girls and young women and adolescent boys and young men in South Africa (SheS’Cap-Linkage). The proposed study is timely and critical in the HIV and AIDS treatment continuum landscape, particularly for South Africa which has the words largest HIV treatment programme. This study seeks to monitor linkages to, and retention in HIV care among adolescent girls and young women (AGYW) and adolescent boys and young men (ABYM) in uMgungundlovu district, KwaZulu-Natal. The study also seeks to study the HIV testing experiences of AGYW and ABYM who have been newly diagnosed with HIV, comparing these experiences across different service delivery models. By doing this, this study will potentially describe the success of different treatment delivery models. My overall impression of the study is that it has been well designed and articulated in the study protocol.

The protocol highlights that there is evidence of adequate literature review that has been consulted in the development of this study, with a clear outline of the research gap that is being addressed. However, there are updated figures that could be used with respect to the 95-95-95 targets with authors quoting 2018 data. I suspect this is from the original protocol developed at the onset of the study.

The study appropriately intends to utilise a mixed methods approach with the purpose of providing evidence that can improve linkages to, and retention in HIV care services for AGYW and ABYM.

The study objectives have been clearly listed and described, and a clear and robust methodology has been provided making it understandable to see how the study objectives will be achieved.

It is difficult to request authors to improve on areas as it would appear that the study is already past the half way point with the investigators committed to these protocols. There is therefore little value in reviewing beyond requesting clarity.

Having said that, perhaps the authors could address and provide clarity in the following areas:

1. The definition of ABYW is not consistent with extent literature, including literature cited in this manuscript. On page 4 second paragraph you refer to adolescents and young people as being between 15-24, including specifically referring to ABYM. To therefore have an expanded definition as reference to your target population is problematic.

2. Your abstract should ideally articulate your primary objective.

3. I would split your objectives into your primary of objective - which is to quantify the linkage and retention rates among HIV positive ABYM & AGYW.

4. Consider just combining stating adolescents and young people (AYP). You can then stratify results by sex.

5. With respect to your section on Setting - why not provide the prevalence of your target population.

6. I think there is a problem with your first stated study question - determining the available service delivery models. Later in the study you have committed to evaluating the three delivery models (facility based, school-based and community based). Does this not render this question redundant?

7. I would strongly encourage the investigators to adopt the language of losses to follow-up (LTFU). We are aware the LTFU might prevent the long-term success of HIV treatment and might delay the achievement of the 95–95-95 objectives. It would have been useful if this study described and analysed of the strategies used to reengage LTFU in HIV care, their implementation and impact. Perhaps beyond the scope of this study but also aligns with the newly published guidelines by WHO (2021). Specifically it states that HIV programmes should implement interventions to trace people who have disengaged from care and provide support for re-engagement. Further, psychosocial interventions should be provided to all adolescents and young adults living with HIV. This study is opportune to evaluate this.

8. Heading: Study Population and recruitment: The penultimate sentence beginning 'This cohort..." needs to be revisited. The word 'collection' may be missing at the end.

9. Given that the study has begun - it may be helpful to add a paragraph describing any deviation from the protocols.

10. The sample size calculation could potentially be problematic. Authors used the LTC rate of 62% without factoring in differences across age cohorts - given that this study is focusing on a particular cohort. Further, as far as I'm aware the LTC rate is drawn from facility based models which I would expect to be higher than school or community based models - where following a positive test the individual will then have to make their way to a clinic. However, it is a conservative estimate give more recent data on a neighboring district.

11. PLOS requires that the data be accessible - either through a data repository or accessible from the PI - this will need to be stated

12. Your listed enablers and barriers appear to be missing a few things which the literature has suggested are important. Many reasons have been given for delayed commencement of ART and these include patient’s choice, prolonged adjustment periods, transport costs due to distance from health facility (be specific about the costs), stigma and fear of disclosure. Other factors associated with delayed ART uptake are staff shortages, long waiting times, fear of drug side effects, and the need to take time off work/school. Furthermore, there are factors that have been associated with delay or lack of linkage to care including lower levels of education, feeling well at diagnosis and alternate healing systems. Not clear that these issues are being analyzed.

13. I would have also included: Baseline clinical characteristics which could have included opportunistic infections, body mass index (BMI), functional status, pregnancy status, any current health complaints, WHO clinical stage and mode of HIV testing (VCT or PICT). Behavioral characteristics such as sero-status of sexual partner, disclosure status, and HIV test history, number of sexual partner/s, condom utilization, and alcohol use. HIV and ART related Knowledge, history of STI. Psychological characteristics could include perceived stigma and psychological distress.

14. With respect to the qualitative work it could have potentially have been enhance by adopting a descriptive phenomenological design. This type of phenomenology places the emphasis on descriptions of personal experiences and delineations of how and what the individual hears, sees, feels, believes, remembers, decide and evaluates. In this particular instance, the search for meaning with regard to the experiences of HIV-positive adults in respect of ART initiation.

15. The authors could provide a justification for sample size, particularly for the qualitative component where it has been stated that 30 participating patients who successfully linked to care, 30 who are not, and 30 who have never tested for HIV will be sampled. It also has not been stipulated whether the proposed sample representative of the target population- this information could be useful. The authors have provided a recruitment plan with a clear inclusion and exclusion criteria, perhaps the authors could provide details on the timelines for their recruitment goal over the 24 months stipulated for data collection.

Finally - wishing the researchers best of luck with this work. I look forward to the results.

Reviewer #2: Overall this is a well written protocol. There are sections a bit repetitive. I have comments around the study design and sample size calculations.

Study design is confusing. For example, the section titled “study design” on page 7, doesn’t mention anything about “clustered” nature of the study. If this is a clustered-design study, then, power/sample size calculations need to account for the design effect which is: 1+ (cluster size-1)*(intra-class correlation coefficient)

In sample-size calculation, there is no information regrading the design effect. In addition, what is the rational for selecting ICC=0.01? selecting such a low ICC indicates almost “no cluster” effect

Study has also a longitudinal aspect. There is no information regarding the “loss-to-follow up”

The investigators randomly mentioned that:

“Primary outcomes will be evaluated using a logistic regression model. A time to event analysis will also be conducted taking the study design into account.” with no further information for the "time to event analysis"

Which outcome(s) will be analysed using a time to event analysis? If there is a time to event data, then survival type of analysis such as Cox regression models should be used but there is no mention in the protocol for such analysis. The investigators only mentioned “logistic regression models” which suits cross-sectional data only. Given the study has clustered-design component, how this will be accounted in logistic regression models?

What are the potential confounders that the investigators are talking about on page 2?

minor comments:

Why the age cut points are for men and women are different? While 15-24 years old

Looking at the Introduction: the vast majority of the background information was referenced using only one reference which is reference #3:

7. PLOS authors have the option to publish the peer review history of their article (what does this mean?). If published, this will include your full peer review and any attached files.

Reviewer #1: **Yes: **Gavin George

Reviewer #2: No

---

## [Author Response · Author response to Decision Letter 0]

28 Oct 2022

The authors would like to thank the reviewers for taken the time to read through this manuscript. Your positive and constructive feedback is much appreciated. The manuscript has been revised accordingly.

EDITOR

1. Your ethics statement should only appear in the Methods section of your manuscript. If your ethics statement is written in any section besides the Methods, please delete it from any other section.

Ethics statement has been deleted from the abstract.

2. We note that Figure 1 in your submission contain image which may be copyrighted. All PLOS content is published under the Creative Commons Attribution License (CC BY 4.0), which means that the manuscript, images, and Supporting Information files will be freely available online, and any third party is permitted to access, download, copy, distribute, and use these materials in any way, even commercially, with proper attribution.

The image in Figure 1 is not copyrighted but the lead author’s work.

Just as observed by the reviewers, the methods is confusing. The authors have to clarify exactly what they will be doing. I suggest the following

Abstract

- present all methods related to the quantitative study first. Present the qualitative methodology afterwards. Please dont mix up the two

- 'The questionnaire study population comprises of 231 healthcare providers and AGYW aged 15-24 years and ABYM aged 15-35 years old.' This is quite confusing because the authors mentioned that they will be interviewing 1100 participants.

Thank you for the suggestions. The abstract has been adjusted accordingly.

“For the quantitative component, a sample of 1100 AGYW aged 15-24 years and ABYM aged 15-35 years old will be recruited into the study, in addition to 231 healthcare providers (HCPs) involved in the implementation of the UTT program. The qualitative component will include 30 participating patients who were successfully linked to care, 30 who were not, and 30 who have never tested for HIV. Key informant interviews will also be conducted with 24 HCPs.” - Page 2.

Introduction

- the authors should include additional references. Ref 3 has been cited too often.

Additional references have been included. Thank you. - Page 4.

Methods

- About how many clients test for HIV at the clinics. The authors should specify, since this will influence the number the participants they will be able to recruit

Thank you. This has been included in the text. 

“The criteria for selecting the 22 SDPs included in this study have been reported elsewhere [19], and the HIV testing number in these SDPs are >4000 clients per month.” - Page 9.

REVIEWER #1

The authors have set out to outline the study protocol entitled Strengthening Health System’s Capacity for Linkage to HIV Care for adolescent girls and young women and adolescent boys and young men in South Africa (SheS’Cap-Linkage). The proposed study is timely and critical in the HIV and AIDS treatment continuum landscape, particularly for South Africa which has the words largest HIV treatment programme. This study seeks to monitor linkages to, and retention in HIV care among adolescent girls and young women (AGYW) and adolescent boys and young men (ABYM) in uMgungundlovu district, KwaZulu-Natal. The study also seeks to study the HIV testing experiences of AGYW and ABYM who have been newly diagnosed with HIV, comparing these experiences across different service delivery models. By doing this, this study will potentially describe the success of different treatment delivery models. My overall impression of the study is that it has been well designed and articulated in the study protocol. The protocol highlights that there is evidence of adequate literature review that has been consulted in the development of this study, with a clear outline of the research gap that is being addressed. However, there are updated figures that could be used with respect to the 95-95-95 targets with authors quoting 2018 data. I suspect this is from the original protocol developed at the onset of the study. The study appropriately intends to utilise a mixed methods approach with the purpose of providing evidence that can improve linkages to, and retention in HIV care services for AGYW and ABYM. The study objectives have been clearly listed and described, and a clear and robust methodology has been provided making it understandable to see how the study objectives will be achieved.

Thank you for the compliment and the constructive comments. We have addressed all your comments including editing the manuscript and rectifying the error in the conclusion section. These are in track changes in the revised manuscript. 

It is difficult to request authors to improve on areas as it would appear that the study is already past the half way point with the investigators committed to these protocols. There is therefore little value in reviewing beyond requesting clarity. Having said that, perhaps the authors could address and provide clarity in the following areas:

1. The definition of ABYW is not consistent with extent literature, including literature cited in this manuscript. On page 4 second paragraph you refer to adolescents and young people as being between 15-24, including specifically referring to ABYM. To therefore have an expanded definition as reference to your target population is problematic.

The definition is consistent throughout the documents; however, we have rewritten the definition for more clarity

“Adolescent girls and young women (AGYW) aged 15-24 years and adolescent boys and young men (ABYM) aged 15-34 years represent one of the populations at highest risk for HIV-infection in South Africa.” - Page 1.

2. Your abstract should ideally articulate your primary objective. 

3. I would split your objectives into your primary of objective - which is to quantify the linkage and retention rates among HIV positive ABYM & AGYW.

Thank you for your comment, unfortunately the word count for the abstract (350) will not allow us to include all the four objectives stated on page 6, however, we have rewritten the purpose (aim) of the study to read:

“Thank you. This has been rewritten to read “The aim of this study is to describe the current approaches for engaging AGYW and ABYM in the treatment continuum to generate knowledge that can guide efforts to improve linkage to, and retention in, HIV care among these populations in KwaZulu-Natal, South Africa.” - Page 2.

4. Consider just combining stating adolescents and young people (AYP). You can then stratify results by sex.

We initially considered this option, however, the age groups for both populations are not the same as you would have noticed in the definition on page 1

5. With respect to your section on Setting - why not provide the prevalence of your target population.

The text has been updated accordingly to read:

“This predominantly rural district, with high HIV prevalence of 24% and 37% for people aged between 15–49 years among males and females, respectively [22]” - Page 7.

6. I think there is a problem with your first stated study question - determining the available service delivery models. Later in the study you have committed to evaluating the three delivery models (facility based, school-based and community based). Does this not render this question redundant?

We respectfully disagree with this comment. Yes, we committed to evaluating the three delivery models (facility based, school-based and community based) to determine the available service delivery models that are seeing higher linkage to care, and continuity of care and treatment among AGYW and ABYM. This is important to describe the current approaches used to engage AGYW and ABYM in the treatment continuum to generate knowledge which can be used to provide recommendations on the best practices for linkage to and retention in HIV care services for these age groups. - Page 6

7. I would strongly encourage the investigators to adopt the language of losses to follow-up (LTFU). We are aware the LTFU might prevent the long-term success of HIV treatment and might delay the achievement of the 95–95-95 objectives. It would have been useful if this study described and analysed of the strategies used to reengage LTFU in HIV care, their implementation and impact. Perhaps beyond the scope of this study but also aligns with the newly published guidelines by WHO (2021). 

Specifically it states that HIV programmes should implement interventions to trace people who have disengaged from care and provide support for re-engagement. Further, psychosocial interventions should be provided to all adolescents and young adults living with HIV. This study is opportune to evaluate this.

We totally agree with this comment, and it is an important area to research, however, like the reviewer noted, this is beyond the scope of our study due to limited funding

8. Heading: Study Population and recruitment: The penultimate sentence beginning 'This cohort..." needs to be revisited. The word 'collection' may be missing at the end.

We beg to disagree with this comment. The word “routine data” refer to data from “…facility-based health information systems, routine data from district-level TIER.Net, data from the National Health Laboratory Service (NHLS)…” used to manage HIV care. - Pages 8 and 12.

9. Given that the study has begun - it may be helpful to add a paragraph describing any deviation from the protocols.

There is currently no deviation from the protocol

10. The sample size calculation could potentially be problematic. Authors used the LTC rate of 62% without factoring in differences across age cohorts - given that this study is focusing on a particular cohort. Further, as far as I'm aware the LTC rate is drawn from facility based models which I would expect to be higher than school or community based models - where following a positive test the individual will then have to make their way to a clinic. However, it is a conservative estimate give more recent data on a neighboring district.

Thank you for the comment. This section has been rewritten to account for your concerns as follows:

“The number of clusters (SDPs) needed to detect a coverage difference between the community- and facility-based service delivery models with >80% power is 22 in total, translating to 11 in each model with 50 participants per cluster assuming a intraclass correlation coefficient (ICC) of 0.01 and significance level of 0.05 [20]. The sample and power calculation were done using Stata v14.2 [21]. The study will retain at least 80% power for a 20% dropout at 6 months of follow-up (m=40 per cluster) and the design effect of 1.5 were deemed realistic for the study populations.” - Page 10.

11. PLOS requires that the data be accessible - either through a data repository or accessible from the PI - this will need to be stated.

This is a protocol and not a research paper, hence no generation of data. However, we plan to publish the findings of this study once completed, and the study data will be made available on request. 

12. Your listed enablers and barriers appear to be missing a few things which the literature has suggested are important. Many reasons have been given for delayed commencement of ART and these include patient’s choice, prolonged adjustment periods, transport costs due to distance from health facility (be specific about the costs), stigma and fear of disclosure. Other factors associated with delayed ART uptake are staff shortages, long waiting times, fear of drug side effects, and the need to take time off work/school. Furthermore, there are factors that have been associated with delay or lack of linkage to care including lower levels of education, feeling well at diagnosis and alternate healing systems. Not clear that these issues are being analyzed.

Yes, these factors will be analyzed and are included in the protocol. See Tables 5a and 5b in the initial protocol (now Supplements 1 and 2) - Pages 17, 18 (Now 22).

13. I would have also included: Baseline clinical characteristics which could have included opportunistic infections, body mass index (BMI), functional status, pregnancy status, any current health complaints, WHO clinical stage and mode of HIV testing (VCT or PICT). Behavioral characteristics such as sero-status of sexual partner, disclosure status, and HIV test history, number of sexual partner/s, condom utilization, and alcohol use. HIV and ART related Knowledge, history of STI. Psychological characteristics could include perceived stigma and psychological distress.

Yes, these factors will be analyzed as part of objectives 1 to 4, and are included in the quantitative questionnaire (Table 4) - Pages 12 and 13.

14. With respect to the qualitative work it could have potentially have been enhance by adopting a descriptive phenomenological design. This type of phenomenology places the emphasis on descriptions of personal experiences and delineations of how and what the individual hears, sees, feels, believes, remembers, decide and evaluates. In this particular instance, the search for meaning with regard to the experiences of HIV-positive adults in respect of ART initiation.

Thank you for this suggestion. It is a good method to incorporate, however, data collection for this component of the study has begun. Including the suggested methodology now may jeopardize the validity of the findings, since it will not be applicable to the rest of the data that have already been collected.

15. The authors could provide a justification for sample size, particularly for the qualitative component where it has been stated that 30 participating patients who successfully linked to care, 30 who are not, and 30 who have never tested for HIV will be sampled. It also has not been stipulated whether the proposed sample representative of the target population- this information could be useful. The authors have provided a recruitment plan with a clear inclusion and exclusion criteria, perhaps the authors could provide details on the timelines for their recruitment goal over the 24 months stipulated for data collection.

“For the qualitative component, 90 purposively selected participants, aggregated by age and sex, who had enrolled at one of the SDPs and a fraction of those who have never had an HIV test will be interviewed. Based on the 1-month follow-up outcome data, 30 participants who were successfully linked to care, 30 who did not and 30 who have never tested for HIV will be included.”

Finally - wishing the researchers best of luck with this work. I look forward to the results.

Thank you.

Reviewer #2

Overall this is a well written protocol. There are sections a bit repetitive. I have comments around the study design and sample size calculations.

Study design is confusing. For example, the section titled “study design” on page 7, doesn’t mention anything about “clustered” nature of the study. If this is a clustered-design study, then, power/sample size calculations need to account for the design effect which is: 1+ (cluster size-1)*(intra-class correlation coefficient)

Thank you for your constructive comment. This has been updated as follows:

“The number of clusters (SDPs) needed to detect a coverage difference between the community- and facility-based service delivery models with >80% power is 22 in total, translating to 11 in each model with 50 participants per cluster assuming a intraclass correlation coefficient (ICC) of 0.01 and significance level of 0.05 [20]. The sample and power calculation were done using Stata v14.2 [21]. The study will retain at least 80% power for a 20% dropout at 6 months of follow-up (m=40 per cluster) and the design effect of 1.5 were deemed realistic for the study populations.” - Page 10.

In sample-size calculation, there is no information regrading the design effect. In addition, what is the rational for selecting ICC=0.01? selecting such a low ICC indicates almost “no cluster” effect

Thank you for this observation. Statement on the design effect has now been include.

“The study will retain at least 80% power for a 20% dropout at 6 months of follow-up (m=40 per cluster) and the design effect of 1.5 were deemed realistic for the study populations.” - Page 10.

Study has also a longitudinal aspect. There is no information regarding the “loss-to-follow up”

“This cohort will be recruited and followed up at one month using questionnaires and at 6 months through routine data.” - Page 8

“Participants will be followed up at 1 and 6 months after enrolment using quantitative and qualitative data collection methods. Data on the HIV testing experience of AGYW and ABYM will be collected cross-sectionally, whereas the linkage and retention data will be monitored prospectively over six months using routine data.” - page 9.

“Secondary data sources, including routine data from facility-based health information systems, routine data from district-level TIER.Net, data from the National Health Laboratory Service (NHLS) and data from the South African Medical Research Council (SAMRC)’s Rapid Mortality Surveillance (RMS) containing monthly information about deaths registered by the South African Department of Home Affairs will be used to obtain information on linkage to and retention in care in the SDPs where this project will be implemented. The RMS, therefore, will be used to confirm that the individual considered as not linked to care or not retained in care was not deceased.” - page 12

The investigators randomly mentioned that:

“Primary outcomes will be evaluated using a logistic regression model. A time to event analysis will also be conducted taking the study design into account.” with no further information for the "time to event analysis"

Information on the "time to event analysis" has been added to the text:

“Given the small number of clusters the degrees of freedom will be adjusted using the Between-Within method recommended by Li & Redden [29] A time to event analysis will also be conducted for the time to ART initiation from the date of testing HIV positive, using a Cox regression model and adjusting the standard errors of the hazard ratio for the clustering of participants within SDPs. Demographic variables such as age and gender as well as barriers and facilitators of linkage to care will also be considered in the modelling process.” - Pages 15 and 16.

Which outcome(s) will be analysed using a time to event analysis? If there is a time to event data, then survival type of analysis such as Cox regression models should be used but there is no mention in the protocol for such analysis. The investigators only mentioned “logistic regression models” which suits cross-sectional data only. Given the study has clustered-design component, how this will be accounted in logistic regression models?

Thank you for the observations. The text has been rewritten as follow:

“Given the small number of clusters the degrees of freedom will be adjusted using the Between-Within method recommended by Li & Redden [29]. A time to event analysis will also be conducted for the time to ART initiation from the date of testing HIV positive, using a Cox regression model and adjusting the standard errors of the hazard ratio for the clustering of participants within SDPs. Demographic variables such as age and gender as well as barriers and facilitators of linkage to care will also be considered in the modelling process.” - Page 16.

What are the potential confounders that the investigators are talking about on page 2?

This has been rewritten - Page 2.

minor comments:

Why the age cut points are for men and women are different? While 15-24 years old

Studies have shown that older men are involved in sexual relationships with younger women and the age-disparate relationships may increase HIV risk for young women. Hence the focus to include men up to 35 years old.

Maughan-Brown B, Evans M, George G. Sexual behaviour of men and women within age-disparate partnerships in South Africa: implications for young women's HIV risk. PLoS One. 2016;11(8):e0159162.

Maughan-Brown B, Kenyon C, Lurie MN. Partner age differences and concurrency in South Africa: implications for HIV-infection risk among young women. AIDS Behav. 2014;18(12):2469–76

Pettifor A, Rees H, Kleinschmidt I, et al. Young people's sexual health in South Africa: HIV prevalence and sexual behaviors from a nationally representative household survey. AIDS. 2005;19:1525–1534. 

Looking at the Introduction: the vast majority of the background information was referenced using only one reference which is reference #3:

Additional references have been included. Thank you. - Page 4.

---

## [Editor Report · Decision Letter 1]

23 Jan 2023

Strengthening Health System’s Capacity for Linkage to HIV Care for adolescent girls and young women and adolescent boys and young men in South Africa (SheS’Cap-Linkage): Protocol for a mixed methods study in KwaZulu-Natal, South Africa.

PONE-D-22-19444R1

Dear Dr. Nicol,

We’re pleased to inform you that your manuscript has been judged scientifically suitable for publication and will be formally accepted for publication once it meets all outstanding technical requirements.

Kind regards,

Sandra Boatemaa Kushitor, Ph.D.

Academic Editor

PLOS ONE
---

## [Editor Report · Acceptance letter]

2 Feb 2023

PONE-D-22-19444R1 

Strengthening Health System’s Capacity for Linkage to HIV Care for adolescent girls and young women and adolescent boys and young men in South Africa (SheS’Cap-Linkage): Protocol for a mixed methods study in KwaZulu-Natal, South Africa. 

Dear Dr. Nicol:

I'm pleased to inform you that your manuscript has been deemed suitable for publication in PLOS ONE. Congratulations! Your manuscript is now with our production department. 

Kind regards, 

on behalf of

Dr. Sandra Boatemaa Kushitor 

Academic Editor

PLOS ONE